# Enhancing Weakly Supervised 3D Medical Image Segmentation through Probabilistic-aware Learning

## Abstract

3D medical image segmentation is a challenging task with crucial implications for disease diagnosis and treatment planning. Recent advances in deep learning have significantly enhanced fully supervised medical image segmentation. However, this approach heavily relies on labor-intensive and time-consuming fully annotated ground-truth labels, particularly for 3D volumes. To overcome this limitation, we propose an innovative probabilistic-aware weakly supervised learning pipeline tailored for 3D medical image segmentation. Our pipeline consists of three key components. Firstly, we introduce a Probability-based Pseudo Label Generation scheme that synthesizes dense 3D segmentation masks from sparsely annotated point annotations. Secondly, we develop a Probabilistic Multi-head Self-Attention network to extract robust probability-driven features, forming the foundation of our Probabilistic Transformer Network. Lastly, we incorporate a Probability-informed Segmentation Loss Function that effectively guides the training process by incorporating annotation confidence. Experimental results demonstrate significant improvements in weakly supervised segmentation, surpassing state-of-the-art methods.

## 1 Introduction

Medical image segmentation plays a crucial role in refining healthcare systems for accurate disease diagnosis and strategic treatment planning (Chen et al., 2021). It aims to delineate anatomical structures across various imaging modalities, providing essential information for healthcare professionals. Deep learning techniques have had a significant impact on medical image segmentation, as demonstrated by recent studies (Zhang et al., 2021d; Milletari et al., 2016; Wang et al., 2022; Cao et al., 2021). Traditional supervised learning methods based on 2D or 3D "U-shaped" encoder-decoder network architectures, such as U-Net (Ronneberger et al., 2015), have been widely used. However, manual annotation can be time-consuming and resource-intensive (Tajbakhsh et al., 2020), leading researchers to explore alternative strategies to reduce the reliance on extensive labeled data. Various techniques have been employed to address the challenge of limited annotations, including data augmentation (Zhang et al., 2021a; Panfilov et al., 2019; Fu et al., 2018), transfer learning (Ma et al., 2019; Qin, 2019), and domain adaptation (Huo et al., 2018; Chen et al., 2019).

Weakly supervised training methods have also gained attention, where minimal annotations like points and scribbles are used to generate pseudo labels for network training (Li et al., 2022; Bearman et al., 2016; Lin et al., 2016). However, weakly supervised approaches face several challenges. Most existing methods predominantly focus on 2D medical image segmentation, leveraging only weak labels in 2D, thereby neglecting the more encompassing 3D weak annotation. Typically, these methods directly employ sparse weak annotations, such as points or scribbles, during the training phase, potentially resulting in significant information loss. Additionally, the confidence level of the annotator in their annotations is often not taken into consideration, omitting a critical factor in the segmentation process.

To address these challenges, we propose an innovative weakly supervised training pipeline for 3D medical image segmentation, with a focus on incorporating probability as a key factor. Our pipeline is guided by the principle that probability should be integrated throughout the entire training and

inference process of a deep learning model, enabling the extraction of robust and expressive features for segmentation. Specifically, our pipeline consists of three fundamental components. Taking inspiration from the uncertainty model proposed in Gal & Ghahramani (2016), we formulate the process of pseudo annotation generation as assigning confidence to specific voxels and their surrounding regions in a 3D medical image. This leads us to develop a Probability-based Pseudo Label Generation strategy, which converts sparse 3D point labels into dense annotations. Recognizing the inherent variance within classes, we introduce a Probabilistic Multi-head Self-Attention (PMSA) mechanism to mitigate noise present in pseudo labels. Anchored by PMSA, we construct a Probabilistic Transformer Network that enhances segmentation performance by capturing the probabilistic distribution of the input-output mapping. To more intricately capture the underlying probabilistic features of pseudo labels, we devise a Probability-informed Segmentation Loss Function that incorporates annotation confidence. This loss function captures the probabilistic nature of the segmentation task, enabling better alignment with the true segmentation boundaries. Consisting of these three modules, our proposed method takes a probabilistic perspective and comprehensively considers annotation, network structure, and gradient backpropagation. By doing so, our method can effectively generate and utilize "dense" weakly supervised signals, while also providing a mechanism to reduce bias in confidence allocation. This integrated approach ultimately facilitates weakly supervised 3D medical image segmentation with minimal annotation cost.

Solid experiments on the BTCV dataset (Landman et al., 2015), consisting of multi-organ abdominal 3D medical images from the MICCAI 2015 Challenge, substantiate the effectiveness of our proposed method. We observe a significant enhancement in performance over existing state-of-the-art weakly supervised methods, and remarkably, our method achieves comparable results with fully supervised counterparts. The three critical components of our framework: pseudo-label generation, network structure, and loss function, all exhibit significant improvements, collectively contributing to the enhanced accuracy of segmentation in our framework. These results highlight our method's potential as a robust and versatile solution for medical image segmentation in weakly supervised settings.

## 2 RELATED WORK

**Medical Image Segmentation:** Medical image segmentation aims to extract objects of interest from medical images obtained through modalities such as Computed Tomography (CT) and Magnetic Resonance Imaging (MRI). Fully Convolutional Networks (FCN) (Long et al., 2015) and U-Net (Ronneberger et al., 2015) have significantly advanced 2D medical image segmentation. Guan et al. (2019) and Ibtehaz & Rahman (2020) propose modifications to U-Net to improve segmentation accuracy. For 3D volumetric medical image segmentation, Çiçek et al. (2016) introduces a 3D U-Net that handles spatial information from 2D slices, while Milletari et al. (2016) presents V-Net with improved feature extraction and reduced computational costs. However, most of the mentioned methods are fully supervised approaches for 2D medical image segmentation. In this paper, our focus is on weakly supervised 3D medical image segmentation, which offers greater annotation efficiency.

**Weakly Supervised Segmentation:** Weakly supervised learning reduces annotation cost by using sparse annotations instead of fully annotated masks. Weak labels such as bounding boxes (Rother et al., 2004; Dai et al., 2015), scribbles (Lin et al., 2016), and points (Bearman et al., 2016) have been utilized. Zhang et al. (2021b) integrates point-level annotation and sequential patch learning for CT segmentation. Roth et al. (2021) designs a point-based loss function with an attention mechanism. Zou et al. (2020) proposes a well-calibrated pseudo-labeling strategy, while Liu et al. (2022) introduces an informative selection strategy. In contrast, our work proposes a "dense" weak annotation approach from a probabilistic perspective.

**Probabilistic Modeling in Deep Learning:** Probabilistic modeling in deep learning handles uncertainty and provides confidence intervals. Shirakawa et al. (2018) uses a Bernoulli distribution to generate network structures. Choi et al. (2021) estimates a probabilistic distribution using mixture density networks for object detection. Zhang et al. (2021c) introduces Bayesian attention belief networks, while Guo et al. (2022) scales dot-product attentions as Gaussian distributions. Our method is the first probabilistic modeling approach for 3D medical image segmentation, incorporating probability in annotation, network structure, and gradient backpropagation, offering advantages for training and inference.

## 3 METHOD

### 3.1 OVERVIEW

Medical image segmentation, commonly known as semantic segmentation of medical images, is the process of partitioning an image into distinct, non-overlapping regions with unique semantic labels. In the case of 3D medical image segmentation, the objective is to segment a 3D volume into subregions that satisfy the following conditions:

$$I = \bigcup_{i=1}^{k} D_i, \quad D_i \cap D_j = \varnothing, \quad \forall i \neq j, ; i, j \in k \tag{1}$$

Here, $I$ represents the input 3D volume, and $C_1, C_2, ..., C_k$ denotes the set of semantic classes. Each subregion $D_i$ is associated with a specific label $C_i$, ensuring that all pixels within $D_i$ are assigned this label. The advantage of performing segmentation directly on 3D volumes, as opposed to 2D slices, lies in the ability to capture spatial information and preserve volumetric context, leading to more accurate and comprehensive segmentation results. In the context of weak supervision, the model is trained on a training set denoted as $(I_1, L_1), (I_2, L_2), ..., (I_n, L_n)$, where $I_i$ $(i = 1, 2, ..., n)$ represents a 3D medical image, and $L_i$ $(i = 1, 2, ..., n)$ denotes the weak label associated with it. During inference, the model generates dense 3D segments for the input volume, providing detailed information about the spatial semantic distribution of the organs.

Fig. 1 illustrates the overview of our method for solving the weakly supervised 3D medical image segmentation task. In our work, UNETR (Hatamizadeh et al., 2022) is adopted as the baseline network. Then, we propose a probabilistic-aware modeling scheme to modify UNETR into a framework that is tailored for our probabilistic weakly supervised 3D medical image segmentation task. The modification consists of three aspects: 1) A Probability-based Pseudo Label Generation Scheme for generating "dense" weak annotations. 2) A Probabilistic Transformer Network, whose key component is the proposed Gaussian-based Multi-head Self-Attention mechanism, to model a probabilistic-aware network architecture. 3) The final loss function, where the proposed probability-informed segmentation loss function serves as the key component. The details of the network architecture can be viewed in Appendix A.1.

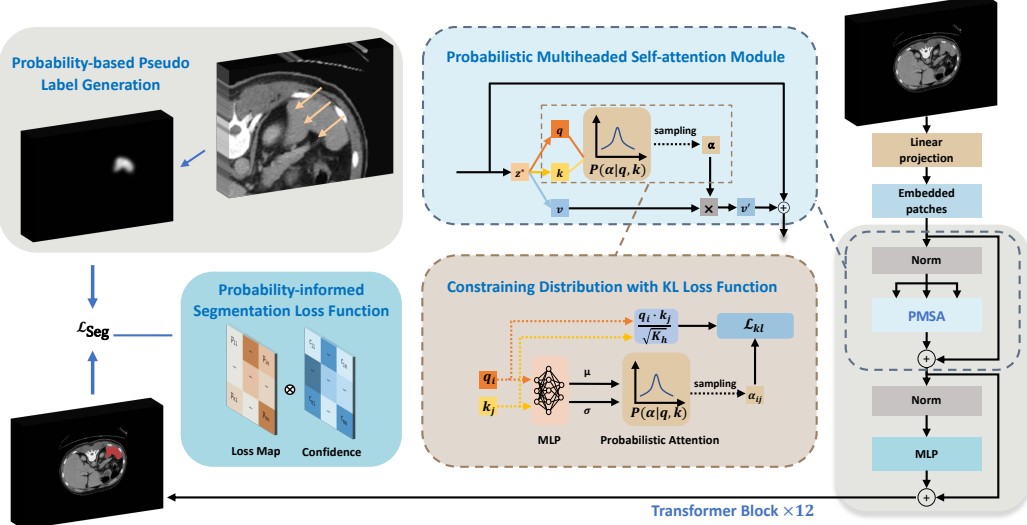

Figure 1: **Overview of our framework.** We adopt UNETR as the baseline network for our segmentation model. The input is a 3D medical volume, which is processed by our Probabilistic Transformer Network, which is powered by the PMSA mechanism. The output of the network is a 3D segmentation map, which is supervised by the "dense" probability-based pseudo label generated from "sparse" point-based annotations. A Probability-aware Segmentation Loss Function is proposed to train the network.

### 3.2 PROBABILITY-BASED PSEUDO LABEL GENERATION SCHEME

#### 3.2.1 SPARSE LABELS ANNOTATION

In this paper, we investigate weakly supervised 3D medical image segmentation as a means to reduce annotation costs. Various annotation methods, such as scribbling and drawing points, can be employed. To facilitate the annotators, we choose 3D points as our sparse labels. In practical applications, we instruct annotators to randomly select 3D points that are evenly distributed on the surface of the target organ, resulting in higher quality pseudo dense labels (See Sec. 3.2.2).

In our experiments, we simulate the aforementioned annotation process to obtain pseudo sparse labels. Firstly, we erode the real ground-truth 3D label using a $5 \times 5 \times 5$ structuring element, denoted as $L' = \text{Erosion}(L, \text{structuring element})$. This operation creates a binary voxel set representing the eroded label. Next, we utilize the Farthest Point Sampling (FPS) algorithm (Eldar et al., 1997) to extract $n$ points within the eroded region, which ensures that the selected points are distributed as evenly as possible. This process can be expressed as $P = \text{FPS}(L', n)$, where $P$ represents the set of 3D points. By employing the erosion operation and Farthest Point Sampling, we can accurately simulate the annotation process and generate pseudo sparse labels that closely approximate the desired distribution on the organ's surface.

#### 3.2.2 PSEUDO LABELS GENERATION

After obtaining the sparse labels, a direct approach would be to use them as supervision signals to calculate losses. However, relying solely on these sparse points leads to significant information loss and may not be sufficient to effectively train a 3D medical image segmentation network. To address this limitation, we propose a dense 3D label generation process. Our key insight is that the process of point annotation assigns confidence scores to the annotated point and its surrounding region, with higher confidence closer to the annotated point. Leveraging this insight, we can easily achieve the generation of dense labels. Specifically, given an annotated point $(x_a, y_a, z_a)$ in a 3D volume, we consider the confidence $P$ at the annotated point $(x_a, y_a, z_a)$ to be the highest, with the confidence of surrounding points $P(x, y, z)$ decreasing as the distance to $(x_a, y_a, z_a)$ increases. To model this behavior, we formulate the confidence score $P(x, y, z)$ as a Gaussian function, centered at $(x_a, y_a, z_a)$, of the coordinates $(x, y, z)$. Mathematically, the confidence score can be expressed as follows:

$$P(x, y, z) = e^{-\frac{(x-x_a)^2 + (y-y_a)^2 + (z-z_a)^2}{2\sigma^2}} \tag{2}$$

Here, $\sigma^2$ is set to 100, and it controls the spread of the Gaussian function. This process is repeated for all $n$ points, and the resulting $n$ label maps are summed together. Finally, the intensity of the label map is normalized to the range $[0, 1]$. We refer to these generated maps as the pseudo dense 3D labels. The entire Probability-based Pseudo Label Generation pipeline is illustrated in Fig. 2 and Appendix A.2.

By employing this Probability-based Pseudo Label Generation process, we effectively convert the sparse annotations into dense 3D labels, capturing the confidence information around each annotated point. This approach enhances the training process and enables more informative supervision for the 3D medical image segmentation network.

### 3.3 PROBABILISTIC TRANSFORMER NETWORK

The proposed pseudo label in Sec. 3.2.2 captures the confidence level of the annotator but exhibits high within-class variance, as illustrated in Fig. 3. This variance arises from the inherent morphological variations of human organs and the randomness of the point-sampling process. To address this, a probabilistic model is needed to capture the complex distribution. Additionally, the confidence of a specific point is correlated with the confidence of its surroundings. Vision-Transformer-based architectures, known for capturing long-range dependencies and global context in images, provide a suitable framework. To this end, we introduce a Probabilistic Transformer network, leveraging the Probabilistic Multi-head Self-Attention (PMSA) component.

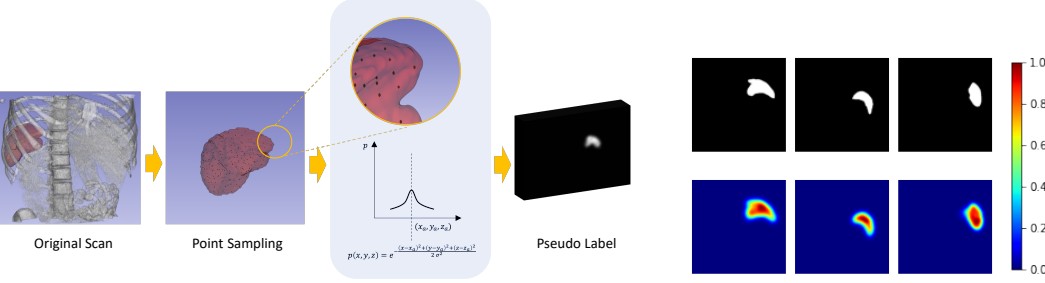

Figure 2: The pipeline for Probability-based Pseudo Label Generation. The points are randomly sampled within the target organ. The Probability-based Pseudo Label is generated by assigning confidence using the sampled points.

Figure 3: The ground truth label and the corresponding Probability-based Pseudo Label of spleen. The three pairs of images belong to different subjects.

### 3.3.1 PROBABILISTIC MULTI-HEAD SELF-ATTENTION COMPONENT

Multi-head Self-Attention (MSA) is a key component in the Transformer model. It captures the dependencies between different positions in an input sequence by using multiple attention heads. In MSA, given an input sequence $z \in \mathbb{R}^{N \times K}$, where $N$ represents the sequence length and $K$ represents the feature dimension at each position, each attention head generates a set of attention weights to compute the attention values for each position with respect to other positions. The calculation of MSA can be expressed as follows:

$$\text{MSA}(z) = \text{Softmax}\left(\frac{QK^T}{\sqrt{d_k}}\right)V \qquad (3)$$

Here, $Q$, $K$, and $V$ are obtained by linearly transforming the input sequence $z$ into query, key, and value representations, respectively. The attention weights are computed by taking the dot product of the query and key vectors, scaled by the square root of the key dimension $d_k$. The softmax function is applied to obtain the final attention weights. Finally, the attention values are computed by multiplying the attention weights with the value vectors.

However, the Probability-based Pseudo Label suffers from large in-class variance caused by the randomness of the point-sampling process and the inherent diversity of human organ structure. To guide our model to capture the variance within the proposed pseudo label and encode the input properly, inspired by Guo et al. (2022), we introduce our Probabilistic Multi-head Self-Attention module. In a single SA head, we assume that the dependency score $\alpha_{ij}$ follows a Gaussian distribution: $\alpha_{ij} \sim \mathcal{N}(\mu_{ij}, \sigma_{ij}^2)$, where the mean $\mu_{ij}$ and the variance $\sigma_{ij}^2$ are calculated with $q_i$ and $k_j$ using a multilayer perceptron (MLP). In order to allow the parameters to be updated through back propagation, we adopt reparameterization trick (Kingma et al., 2015):

$$\alpha_{ij} = \mu_{ij} + \sigma_{ij}\epsilon, \quad \epsilon \sim \mathcal{N}(0,1) \qquad (4)$$

where $\epsilon$ is a random variable that follows a standard normal distribution. For other parameters in the model, we set them as deterministic, and denote them as $\Theta$.

We assume that the dependency scores within the same PMSA layer are independent of each other, while the dependency scores of deeper PMSA layer is dependent on those of former PMSA layers:

$$\alpha_l \sim p(\alpha_l|X',\Theta,\alpha_{l-1}...,\alpha_1), \quad l = 1,...,L \qquad (5)$$

where $\alpha_l$ denotes the dependency scores of the PMSA layer of the $l$th transformer block.

With PMSA, the distribution of the output segmentation map $y'$ given the input image $X'$ can be computed according to:

$$P(y'|X',\Theta) = E_{\alpha \sim p(\alpha|X',\Theta)}[P(y'|X',\Theta,\alpha)] = \int_{\alpha} P(y'|X',\Theta,\alpha)p(\alpha|X',\Theta)d\alpha. \qquad (6)$$

However, due to the intractability of the integral in Eq. (6), we sample $\alpha$ from $p(\alpha|X', \Theta)$ for $M$ times to approximate the integral, in which every $\alpha_{ij}$ is sampled independently each time:

$$y^* = \underset{y'}{\operatorname{argmax}} \sum_{m=1}^{M} \frac{1}{M} P(y'|X', \Theta, \alpha_m) \tag{7}$$

where $\alpha_m$ denotes the dependency scores sampled at the $m$th time, and $y^*$ is the final segmentation output. The proof of Eq. (6) is in Appendix A.3.

### 3.3.2 PROBABILISTIC TRANSFORMER NETWORK ARCHITECTURE POWERED BY PMSA

Our framework follows the contracting-expanding pattern of the UNETR architecture. Initially, a 3D volume $x$ with dimensions $(H, W, D)$ and $C$ input channels is divided into non-overlapping uniform patches. This division results in a sequence $x_v$ by flattening the patches, where $N = (H \times W \times D)/P^3$ represents the length of the sequence. These patches are then projected into a lower-dimensional embedding space using a linear layer. Additionally, a 1D learnable positional embedding $E_{pos}$ is added to the projected patches. The features are then passed through a series of Probabilistic Transformer blocks, which consist of alternating layers of PMSA and MLP (Multi-Layer Perceptron) blocks. These blocks help capture contextual information and refine the features. The output of the Probabilistic Transformer blocks, denoted as $z_i$, has a reduced spatial dimension compared to the input. In the decoder network, the features undergo deconvolution blocks to increase the resolution. Starting from the lowest resolution, the features are concatenated and upsampled to match the resolution of higher-level features. This process is repeated until the full resolution is restored. Finally, the output layer utilizes the feature map with the full resolution to predict the final segmentation results.

### 3.4 PROBABILITY-INFORMED SEGMENTATION LOSS FUNCTION

As discussed in Sec. 3.2, the proposed pseudo label is considered as a probability map, where the intensity of each point represents the annotator's confidence in classifying it as the target organ. Therefore, to enable our model to be aware of the underlying confidence within the pseudo label, we introduce a loss function which is a combination of DICE loss and Probability-weighted Cross Entropy (PCE) loss. The intuition is that points with prior confidence greater than a certain threshold are considered as the foreground of the basic label map, while we weight the loss function with the prior confidence of the annotator since voxels with low confidence deserve lower loss weights.

Given the output $y^*$ and pseudo label map $S$, the segmentation loss is formulated as:

$$\mathcal{L}_{\text{Seg}} = \mathcal{L}_{\text{DICE}}(y^*, S_T) + \mathcal{L}_{\text{PCE}}(y^*, S_T) \tag{8}$$

where $S_T$ is the thresholded map of $S$ with a threshold $T$ (set as 0.5), and for each voxel in the segmentation map, $\mathcal{L}_{\text{pce}}$ is formulated as:

$$\mathcal{L}_{\text{pce}}(p_i, s_i) = \begin{cases} s_i \log(p_i), & \text{if } s_i \geq T \ (1 - s_i) \log(1 - p_i), \\ \text{if } s_i < T \end{cases} \tag{9}$$

where $s_i$ and $p_i$ are the confidence of the $i$th voxel in $S$ and $y^*$, respectively, and $N$ denotes the number of voxels in the segmentation map. The PCE loss is then averaged over all voxels to obtain $\mathcal{L}_{\text{PCE}}$:

$$\mathcal{L}_{\text{PCE}} = \frac{1}{N} \sum i = 1^N \mathcal{L}_{\text{pce}}(p_i, s_i) \tag{10}$$

Moreover, to guide the model in learning a more effective probabilistic representation of features, we introduce Kullback-Leibler (KL) Loss to encourage the distributions of dependency scores to align with a set of predefined Gaussian distributions. This assumption helps ensure that the dependency score distributions are close to certain manually-set prior distributions. The KL loss is defined as:

$$\mathcal{L}_{KL} = \sum_{l=1}^{L} \sum_{i,j} \left( \mathcal{D}_{KL}(p(\alpha_{lhij}|X', \Theta, \alpha_{k-1}, ..., \alpha_1)|\mathcal{N}(\alpha'_{lhij}, \sigma^2))) \right) \tag{11}$$

Here, $l$ and $h$ denote the in DICE of the transformer block and head, respectively. $\alpha_{lhij}$ represents the dependency score sampled from the distribution $\mathcal{N}(\mu_{lhij}, \sigma^2_{lhij})$, while $\alpha'_{lhij}$ is calculated as the scaled dot-product of $q_{lhi}$ and $k_{lhj}$. By minimizing the KL loss, we encourage the distribution $\mathcal{N}(\mu_{lhij}, \sigma^2_{lhij})$ to closely match $\mathcal{N}(\mu'_{lhij}, \sigma^2)$, where $\sigma$ is empirically set to 1.

The overall probability-aware segmentation loss function is formulated as:

$$\mathcal{L}_{total} = \mathcal{L}_{Seg} + w\mathcal{L}_{KL}. \tag{12}$$

where $w$ is a balance term to prevent $\mathcal{L}_{KL}$ from dominating the update of parameters through back-propagation, empirically set as 0.3.

## 4 EXPERIMENTS

### 4.1 IMPLEMENTATION DETAILS

We conduct experiments on BTCV dataset (Landman et al., 2015), which comprises 30 multi-organ abdominal 3D CT scans acquired during portal venous contrast phase. The experiments are conducted on a single NVIDIA RTXA5000 GPU with 24GB GPU memory. Following the approach in Hatamizadeh et al. (2022), we set the number of transformer encoders to 12 (L=12) with an embedding size of 768 (K=768). Each patch has a resolution of 16x16x16. During training, we use the AdamW optimizer with an initial learning rate of 0.0001 and a batch size of 1. The number of training iterations was set to 6,000. For inference, we employ a sliding window approach with a 50% overlap. The number of sampled points for different labels is proportional to the volume of the corresponding organ: 200 points for the spleen, 400 points for the liver, and 50 points for each of the right and left kidneys. We assess the efficacy of the methodology using two prevalent metrics, the DICE score (DICE), where higher is better, and the 95% Hausdorff Distance (HD95), where lower is better. For more details, please refer to Appendix A.4.

### 4.2 RESULTS

**Comparison with SOTA weakly supervised methods.** Tab. 1 delineates a comparison of our proposed method against other cutting-edge weakly supervised methods, such as sparse slices annotation (Çiçek et al., 2016), convex shapes via quickhull (Barber et al., 1996), and superpixel-based self-supervised learning (SSL) (Ouyang et al., 2022), evaluating DICE and HD95 across four organs. Our method consistently achieves the highest DICE scores, revealing superior segmentation accuracy across all organs. Although our method doesn't always secure the lowest HD95 values, it still maintains competitive, with notable figures being 63.09 for the spleen and 126.22 for the right kidney. Self-supervised learning (SSL), despite its weakly supervised classification, leverages full annotations with precise boundaries in segmentation tasks, giving it an apparent advantage in the HD95 metric. Nevertheless, our method, devoid of such detailed boundary annotations during training, still demonstrates exceptional prowess, surpassing Sparse and Convex methods and underlining its unique capability in addressing weakly supervised 3D medical image segmentation tasks.

In conclusion, our method exemplifies remarkable superiority, consistently securing higher DICE scores and competitive HD95 values, showcasing more accurate and meticulous organ segments in comparison to other methods. The quantitative comparison in Fig. 4 further accentuates our method's proficiency in acquiring more accurate and comprehensive segments.

Table 1: Comparison with SOTA weakly supervised methods.

| Method | Spleen | | Liver | | Left Kidney | | Right Kidney | |
|---|---|---|---|---|---|---|---|---|
| | DICE | HD95 | DICE | HD95 | DICE | HD95 | DICE | HD95 |
| Sparse (Çiçek et al., 2016) | 0.5515 | 116.74 | 0.4303 | 183.29 | 0.2532 | 168.93 | 0.2703 | 123.33 |
| Convex (Barber et al., 1996) | 0.8232 | 69.65 | 0.6268 | 311.14 | 0.4037 | 351.35 | 0.3272 | 325.52 |
| SSL (Ouyang et al., 2022) | 0.7455 | 21.52 | 0.7916 | 46.16 | 0.594 | 29.03 | 0.535 | 40.11 |
| **Ours** | **0.8279** | **63.09** | **0.8157** | **265.79** | **0.7599** | **266.17** | **0.7164** | **126.22** |

Figure 4: Qualitative comparison with weakly supervised methods.

**Comparison with SOTA fully supervised methods.** To underscore the efficacy of our proposed approach, we juxtapose our weakly supervised method against state-of-the-art fully supervised methods, namely TransUnet (Chen et al., 2021), SwinUnet (Cao et al., 2021), and UCTransNet (Wang et al., 2022). It is paramount to note that this juxtaposition is inherently imbalanced, as our method operates on notably sparser original annotations compared to the comprehensive annotations utilized by the aforementioned fully supervised methods. Despite this inherent disparity, as delineated in Table Tab. 2, our method exhibits performances that are remarkably on par with, and in certain metrics, even surpass, those achieved by fully supervised counterparts. For instance, our method eclipses UCTransNet in spleen segmentation, showcasing the distinct advantages of our probabilistic weakly supervised approach. We present visual illustrations of varied experimental results on four organs, refer to Appendix A.5.

In conclusion, our method demonstrates its prowess and superior adaptability, ensuring commendable accuracy even with limited annotations and emphasizing its potential as a robust solution in the realm of medical image segmentation.

Table 2: Comparison with SOTA fully supervised methods.

| Method | | Spleen | | Liver | | Left Kidney | | Right Kidney | |
|---|---|---|---|---|---|---|---|---|---|
| | | DICE | HD95 | DICE | HD95 | DICE | HD95 | DICE | HD95 |
| Fully | TransUnet | 0.8697 | 30.14 | 0.9341 | 10.21 | 0.7822 | 28.19 | 0.8431 | 29.24 |
| | SwinUnet | 0.8294 | 27.38 | 0.9129 | 13.50 | 0.8017 | 63.74 | 0.801 | 28.12 |
| | UCTransNet | 0.8176 | 29.22 | 0.8972 | 17.36 | 0.7822 | 22.77 | 0.7805 | 27.71 |
| | UNETR | 0.9304 | 18.65 | 0.9017 | 39.26 | 0.9159 | 51.00 | 0.8945 | 6.35 |
| Weakly | **Ours** | **0.8279** | **63.09** | **0.8157** | **265.79** | **0.7599** | **266.17** | **0.7164** | **126.22** |

### 4.3 ABLATION STUDY

We integrate a probabilistic mechanism within our model across three critical parts: Pseudo label generation, network structure, and loss function. As delineated in Tab. 3, our primary focus in this section is to examine the efficacy of these dimensions on the BTCV dataset, utilizing the DICE score metric as our evaluation metric. For additional ablation studies addressing crucial parameters such as the number of sampled points and the selection of variance, kindly refer to Appendix A.6.

**PART I. Effectiveness of Probability-based Pseudo Label Generation**. We first investigate the probabilistic mechanism in our weakly pseudo label generation scheme. The experimental results are presented in Part I of Tab. 3. In this table, the "Sparse" column represents our method using sparse 3D points as pseudo labels, while the "Dense" column indicates the conversion of sparse 3D points into dense pseudo labels without considering annotation confidence using a probability mechanism. From the experimental results, it is evident that our method consistently outperforms both the "Sparse" and "Dense" methods for every studied organ. This demonstrates the enhanced accuracy and reliability of our method in weakly supervised 3D medical images segmentation.

**PART II. Effectiveness of Probabilistic Transformer Network Structure**. We then investigate the impact of the probabilistic mechanism in the network architecture. Part II of Tab. 3 presents our experimental results, comparing the performance of the Probabilistic Multi-head Self-Attention (PMSA) and the Multi-head Self-Attention (MSA) methods. The results demonstrate the enhanced precision of PMSA in anatomical structure segmentation, surpassing MSA for all tested organs. These results indicate the heightened accuracy and reliability of PMSA in producing segmentation results that closely align with the actual anatomical structures, demonstrating the significance of considering probabilistic modeling in our transformer network.

**PART III. Effectiveness of Probability-informed Segmentation Loss Function**. Finally, we examine the effectiveness of our designed loss function, as presented in Part III of Tab. 3. The conclusive results clearly highlight the superior performance of our method. Our approach consistently achieves higher effectiveness scores, demonstrating its ability to deliver more accurate and coherent segments. Compared to our approach, the existing non-probabilistic loss functions, specifically DICE, Cross-Entropy (CE), combined Dice-Cross-Entropy (DCE), and Focal, demonstrate suboptimal performance, especially in segmenting the liver and both kidneys. These findings underscore the limitations of the existing loss functions and underscore the superiority of our designed probability-informed loss function in achieving improved 3D medical image segmentation results.

Table 3: Quantitative results of ablation study. For each part of comparison, only the methods corresponding to that part are varied, while the methods in the other parts are kept constant, and the "Ours" column represents the results of our complete approach.

| | PART I | | PART II | PART III | | | | Ours |
|---|---|---|---|---|---|---|---|---|
| | Sparse | Dense | MSA | DICE | CE | DCE | Focal | |
| Spleen | 0.0003 | 0.0002 | 0.817 | 0.3561 | 0.6341 | 0.7665 | 0.7853 | **0.8279** |
| Liver | NA | NA | 0.7719 | 0.5992 | 0.7767 | 0.7753 | 0.4025 | **0.8157** |
| Left Kidney | 0.0002 | 0.0004 | 0.4252 | 0.2599 | 0.3963 | 0.6112 | 0.5653 | **0.7599** |
| Right Kidney | 0.0003 | 0.0002 | 0.5445 | 0.3403 | 0.4839 | 0.506 | 0.3675 | **0.7164** |

## 5 CONCLUSION

In this work, we propose a novel probability-based segmentation regime in weakly supervised 3D medical image segmentation setting. We specifically propose the following enhancements: 1) A novel Probability-based Pseudo Label Generation Scheme to generate pseudo dense 3D labels from points annotations. 2) A novel Probabilistic Transformer Network that benefits most from the proposed Gaussian-based Multi-head Self-Attention Module to learn novel features. 3) A Probability-informed Segmentation Loss Function to supervise the training process. Experimental results demonstrate the superiority of our proposed method.

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

# A APPENDIX

## A.1 DETAILED NETWORK ARCHITECTURE

Our framework follows the contracting-expanding pattern of the UNETR architecture. Initially, a 3D volume $x \in \mathbb{R}^{H \times W \times D \times C}$ with dimensions $(H, W, D)$ and $C$ input channels is divided into non-overlapping uniform patches with dimensions $(P, P, P)$. This division results in a sequence $x_v \in \mathbb{R}^{N \times (P^3 \cdot C)}$ by flattening the patches, where $N = (H \times W \times D)/P^3$ represents the length of the sequence. These patches are then projected into a $K$-dimensional embedding space using a linear layer. Additionally, a 1D learnable positional embedding $E_{pos} \in \mathbb{R}^{N \times K}$ is added to the projected patches. The process can be defined as follows:

$$z_0 = [x_v^1 E; x_v^2 E; ...; x_v^N E] + E_{pos} \tag{13}$$

Here, $E \in \mathbb{R}^{(P^3 \cdot C) \times K}$ represents the patch embedding projection.

The features are then passed through a series of Probabilistic Transformer blocks, which consist of alternating layers of PMSA and MLP blocks. The equations for these blocks are as follows:

$$z_i' = \text{PMSA}(\text{Norm}(z_{i-1})) + z_{i-1}, \quad i = 1...L \tag{14}$$

$$z_i = \text{MLP}(\text{Norm}(z_i')) + z_i', \quad i = 1...L \tag{15}$$

The output of the Probabilistic Transformer blocks, denoted as $z_i$ (where $i$ takes values 3, 6, 9, 12), has a shape of $\frac{H \times W \times D}{P^3} \times K$ and is reshaped into $\frac{H}{P} \times \frac{W}{P} \times \frac{D}{P} \times K$.

In the decoder network, each feature $z_i$ undergoes deconvolution blocks to increase the resolution by a specific factor (2 for $z_{12}$ and $z_9$, 4 for $z_6$, and 8 for $z_3$). Starting from the lowest resolution, i.e., $z_{12}$ and $z_9$, the features are concatenated and upsampled to match the resolution of higher-level features. This process is repeated until the full resolution is restored. Finally, the output layer utilizes the feature map with the full resolution to predict the final segmentation results.

## A.2 THE ALGORITHM OF TARGET GENERATION

The core idea of this algorithm is to generate a three-dimensional Gaussian distribution based on the coordinates of each sampled point, and to accumulate these distributions to form the final label map. This label map can subsequently be used for training medical image segmentation models. The variance $\sigma^2$ influences the width of the generated Gaussian distribution, thereby altering the shape of the label map. This is verified in Appendix A.6.2.

---

**Algorithm 1** Target Generation

---

**Input:** $C = \{x_i, y_i, z_i\}_{i=1}^n$: coordinates of sampled points, $X, Y, Z$
**Output:** $P \in \mathbb{R}^{X \times Y \times Z}$: label map
**for** $\{x_i, y_i, z_i\}$ *in* $C$ **do**
    $P_i \leftarrow 0$ **for** $x \leftarrow 0$ *to* $X$ **do**
        **for** $y \leftarrow 0$ *to* $Y$ **do**
            **for** $z \leftarrow 0$ *to* $Z$ **do**
                $P_i(x, y, z) = e^{-\frac{(x-x_i)^2 + (y-y_i)^2 + (z-z_i)^2}{2\sigma^2}}$
            **end**
        **end**
    **end**
**end**
$P = \sum_{i=1}^n P_i$ Normalize $P$ to $[0, 1]$

---

A.3 PROOF OF EQ. (6)

For the decoder side, a $2 \times 2 \times 2$ deconvolutional layer is positioned at the last output layer $z_{12}$ of the Probabilistic Transformer blocks. The resolution of $z_{12}$ is increased by a factor of 2. Then it is concatenated with the former transformer output, $z_9$, and fed into consecutive $3 \times 3 \times 3$ convolutional layers before another upsample operation. The process is repeated for four times to obtain the original input resolution. The final $1 \times 1 \times 1$ convolutional layer with a sigmoid activation function is adopted to produce the 3D volumetric segmentation prediction. During inference, we sample all the dependency scores independently for $M$ times and calculate the final segmentation output $y^*$ where $\alpha_m$ denotes the dependency scores sampled at the $m$th time.

The proof of Eq. (6) is established as: Given that the dependency scores within the same PMSA layer are independent of each other, and the dependency scores of deeper PMSA layer are dependent on those of former PMSA layers, could be written as:

$$
\begin{aligned}
P(y'|X', \Theta) &= \int_{\alpha} P(y'|X', \Theta, \alpha) p(\alpha|X', \Theta) d\alpha \\
&= \int_{\alpha_1} ... \int_{\alpha_L} P(y'|X', \Theta, \alpha_1, ..., \alpha_L) p(\alpha_L \\
&\quad |X', \Theta, \alpha_1, ..., \alpha_{L-1}) d\alpha_L ... p(\alpha_1|X', \Theta) d\alpha_1 \\
&\approx \int_{\alpha_1} ... \int_{\alpha_{L-1}} \frac{1}{M_L} \sum_{m_L=1}^{M_L} P(y'|X', \Theta, \alpha_1, ..., \\
&\quad \alpha_{Lm_L}) p(\alpha_{L-1}|X', \Theta, \alpha_1, ..., \alpha_{L-2}) d\alpha_{L-1} \\
&\quad ... p(\alpha_1|X', \Theta) d\alpha_1 \\
&\approx \frac{1}{M_1} \sum_{m_1=1}^{M_1} ... \frac{1}{M_L} \sum_{m_L=1}^{M_L} P(y'|X', \Theta, \alpha_{1m_1}, ..., \\
&\quad \alpha_{Lm_L}) \\
&\approx \frac{1}{M} \sum_{m_1=1}^{M_1} ... \sum_{m_L=1}^{M_L} P(y'|X', \Theta, \alpha_{1m_1}, ..., \\
&\quad \alpha_{Lm_L}) \\
&\approx \frac{1}{M} \sum_{m=1}^{M} P(y'|X', \Theta, \alpha_{1m}, ..., \alpha_{Lm})
\end{aligned}
\tag{16}
$$

where $M = \prod_{l=1}^{L} M_l$, which we empirically set as 6 in our experiments.

A.4 THE DETAILS OF IMPLEMENTATION

**BTCV dataset (Landman et al., 2015)**: The MICCAI 2015 Multi-Atlas Abdomen Labeling Challenge dataset consists of 30 3D CT scans of the abdomen acquired during the portal venous contrast phase. Each CT scan contains 85 to 198 slices with voxel dimensions of $512 \times 512$ and in-plane resolution ranging from $0.54 \times 0.54mm^2$ to $0.98 \times 0.98mm^2$. The slice thickness varies from 1 to 6 mm. The dataset was meticulously labeled by trained annotators and reviewed by professional radiologists or radiation oncologists to ensure accuracy. During preprocessing, the images are resampled to have isotropic voxels with a voxel spacing of $1.0mm$. Additionally, the intensity of each image is independently normalized to the range [0, 1]. Following the approach in (Hatamizadeh et al., 2022), the input images are randomly sampled to a size of [96, 96, 96]. The foreground and background samples are balanced with a ratio of 1:1. The dataset is randomly divided into 20 cases for training and 10 cases for validation. The average DICE score, ranging from 0 to 100 (0 for a mismatch and 100 for a perfect match), is reported on four organs (spleen, liver, right kidney, and left kidney) using the validation set.

**Implementation:** We implemented our method using the PyTorch (Paszke et al., 2019) framework and MONAI[1]. All experiments were conducted on a single NVIDIA RTXA5000 GPU with 24GB GPU memory. Following the configuration in (Hatamizadeh et al., 2022), we set the number of transformer encoders to $L = 12$, with an embedding size of $K = 768$. Each patch has a resolution of $16 \times 16 \times 16$. During training, we used the AdamW optimizer (Loshchilov & Hutter, 2017) with an initial learning rate of 0.0001 and a batch size of 1. The models were trained for 6,000 iterations. During inference, we employed a sliding window approach (Keogh et al., 2001) with a 50% overlap. The number of sampled points for each label was proportional to the volume of the corresponding organ (200 for the spleen, 400 for the liver, and 50 for the right and left kidneys).

**Metric:** We evaluate the results using two widely adopted metrics: the DICE score and the 95% Hausdorff Distance. The DICE score is commonly used to measure the similarity between the predicted segmentation region and the ground truth. It calculates the overlap between the two regions and provides a value between 0 and 1. The HD95 metric measures the maximum discrepancy between the boundaries of the predicted segmentation and the ground truth. It quantifies the largest distance between corresponding points on the boundaries of the two regions. A smaller HD95 value indicates that the predicted segmentation is closer to the ground truth, suggesting a better alignment between the boundaries.

## A.5 MORE QUALITATIVE RESULTS

In Fig. 5 and Fig. 6, we present a comparison of our approach with several fully supervised methods and additionally display some visual results to illustrate that our method attains performance comparable to that of fully supervised ones. The visual comparisons provide compelling evidence of the effectiveness of our method in accurately segmenting the desired regions of interest. Despite being trained with limited supervision, our approach demonstrates competitive performance, highlighting its potential as a viable alternative to fully supervised methods.

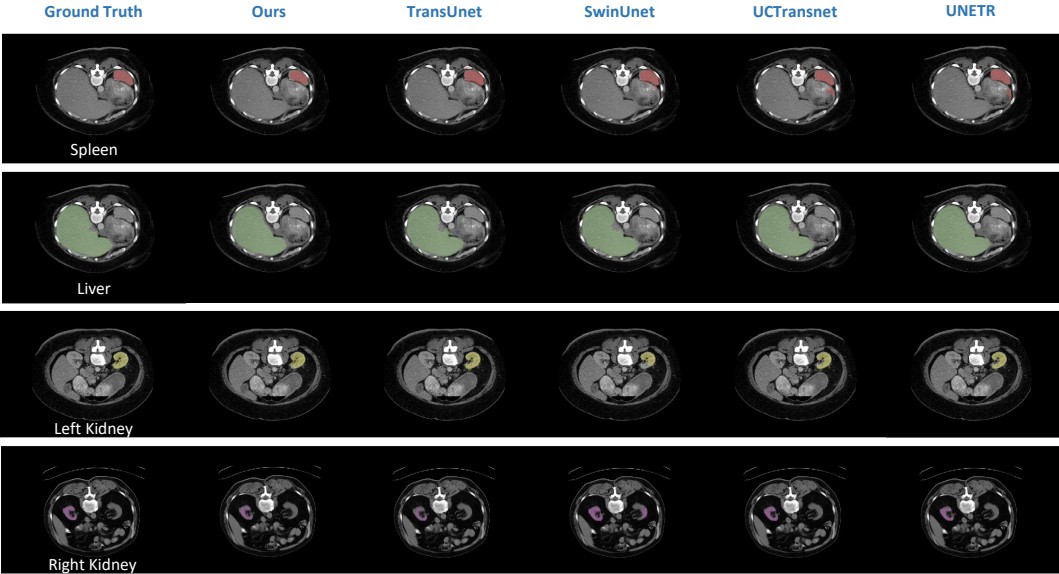

Figure 5: Qualitative comparison with fully-supervised methods. Our method achieves comparable performance with fully-supervised methods.

---

[1]https://monai.io/

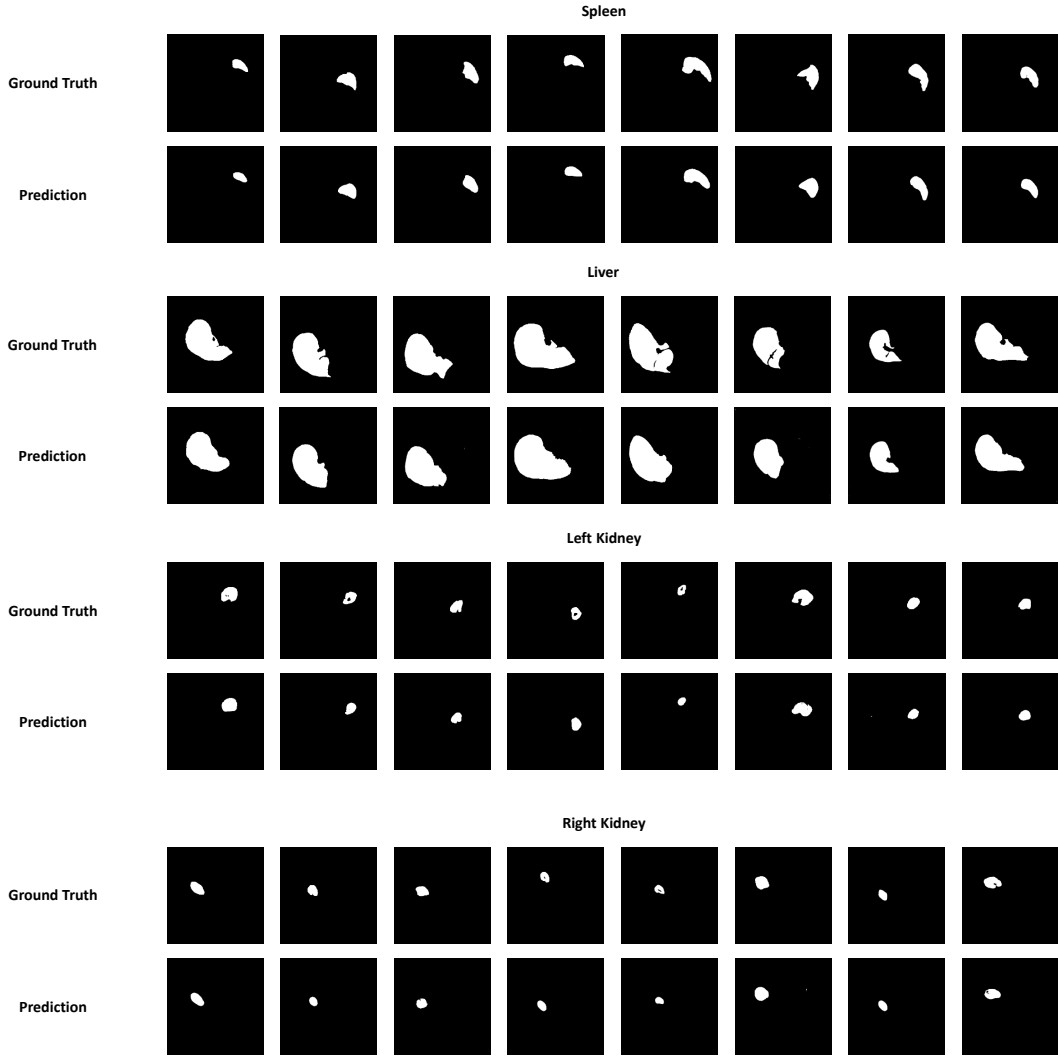

Figure 6: Qualitative results of segmentation prediction discrepancies between ground truth and prediction across four organs.

## A.6 MORE ABLATION STUDIES

### A.6.1 IMPACT OF THE NUMBER OF SAMPLED POINTS

We outline our annotation strategy in Sec. 3.2.1, which involves determining the number of sampled points. In Tab. 4, we investigate the relationship between the number of sampled points and the final segmentation results. Specifically, we keep all other parameters constant and vary only the number of sampled points for pseudo label generation in a comparative experiment. The results demonstrate that the number of sampled points is directly proportional to the segmentation performance, aligning with our expectation that a higher number of points provides better knowledge about the region of interest. While fully supervised segmentation can be seen as providing prior knowledge about all points in the semantic space to the model, our approach strikes a balance between annotation efficiency and segmentation effectiveness. By leveraging a limited number of sampled points, our method maximizes the utilization of available annotation resources while still achieving competitive segmentation results. This annotation strategy allows us to efficiently train the model and obtain accurate segmentations without relying on full supervision.

Table 4: Illustration of the number of sampled points. The horizontal coordinates are the number of sampled points, and the vertical coordinates on the left and right sides are DICE and HD95, respectively. Sampling 200 points achieves the best DICE and HD95.

| n | Spleen | | Liver | | Left Kidney | | Right Kidney | |
|---|---|---|---|---|---|---|---|---|
| | DICE | HD95 | DICE | HD95 | DICE | HD95 | DICE | HD95 |
| 50 | 0.6392 | 336.25 | 0.1081 | 127.16 | 0.6276 | 135.88 | 0.5678 | 292.41 |
| 100 | 0.8001 | 174.41 | 0.7307 | 295.54 | 0.6366 | 197.66 | 0.4772 | 312.45 |
| 150 | 0.5462 | 348.11 | 0.7164 | 304.69 | 0.3856 | 333.5 | 0.5756 | 183.65 |
| 200 | 0.8279 | 63.08 | 0.8157 | 265.79 | 0.7599 | 266.17 | 0.7164 | 126.22 |

### A.6.2   COMPARISON OF THE SELECTION OF VARIANCE $\sigma^2$.

When calculating the KL loss in our probability-informed segmentation loss function, the variance $\sigma^2$ serves as a hyperparameter that needs to be manually determined. To ensure experimental rigor, we investigate the effects of different variances on segmentation accuracy. Table 5 illustrates that the choice of variance in the KL loss significantly influences the final results. We observe that when setting the variance to 1, our model achieves the highest DICE score of 0.8279 and the lowest HD95 value of 63.09. Based on these empirical findings, we establish $\sigma^2$ as 1 in our method. By conducting this analysis, we enhance the reliability of our experimental setup and demonstrate the importance of selecting an appropriate variance for the KL loss. The chosen value of $\sigma^2$ contributes to optimizing the segmentation performance and ensures the robustness of our method.

Table 5: Illustration of selection of variance $\sigma^2$. The horizontal coordinates are the values of $\sigma^2$. The vertical coordinates on the left and right sides are DICE and HD95, respectively. Setting $\sigma^2$ as 1 achieves the best DICE and HD95.

| $\sigma^2$ | Spleen | | Liver | | Left Kidney | | Right Kidney | |
|---|---|---|---|---|---|---|---|---|
| | DICE | HD95 | DICE | HD95 | DICE | HD95 | DICE | HD95 |
| 0.1 | 0.5104 | 394.87 | 0.7478 | 299.2 | 0.5474 | 329.39 | 0.5192 | 273.61 |
| 1 | 0.8279 | 63.09 | 0.8157 | 265.79 | 0.7599 | 266.17 | 0.7164 | 126.22 |
| 10 | 0.5754 | 388.11 | 0.7691 | 312.36 | 0.6103 | 238.93 | 0.3468 | 323.79 |

