# OpenReview forum: "Enhancing Weakly Supervised 3D Medical Image Segmentation through Probabilistic-aware Learning"
_ICLR.cc/2024/Conference — ICLR 2024 Conference Withdrawn Submission_

### Official Review · Reviewer_y2ei · 2023-10-28

**Soundness:** 1 poor
**Presentation:** 2 fair
**Contribution:** 1 poor
**Rating:** 3
**Confidence:** 4

**Summary:**

The paper proposes a probabilistic-aware weakly supervised learning pipeline designed specifically for 3D medical image segmentation. The proposed pipeline consists of three key components: a Probability-based Pseudo Label Generation scheme, a Probabilistic Multi-head Self-Attention network, and a Probability-informed Segmentation Loss Function.

**Strengths:**

The paper presents a probabilistic-aware pipeline tailored for 3D medical image segmentation.

The writing flow is clear.

**Weaknesses:**

For probability-based pseudo label generation scheme part, this work samples 3d point on the surface of the target organ. Instead of sampling, why not directly obtain dense annotations using the organ surface? The segmentation ground truth is nearly identical to the organ surface, making it unreasonable to use weakly supervised learning when ground truth annotations are available.

Sections 3.3 and 3.4 exhibit similarities with VAE at a high level. It is unclear why a probabilistic approach is used instead of a deterministic approach, and further clarification is needed to justify this decision.

The SSL in table 1 is ‘Self- supervised learning for few-shot medical image segmentation’, not weakly supervised method. This discrepancy should be addressed and corrected.

The experiments conducted only on the BTCV dataset, which is relatively small, raises concerns about the generalizability of the proposed method. Considering the availability of large CT datasets, it is essential to investigate datasets with a substantial amount of CT scans to ensure the scalability and applicability of the proposed pipeline, such as AMOS, WORD and AbdomenAtlas-8K.

**Questions:**

See Weaknesses part.

---

### Official Review · Reviewer_eHUZ · 2023-10-30

**Soundness:** 2 fair
**Presentation:** 3 good
**Contribution:** 2 fair
**Rating:** 3
**Confidence:** 5

**Summary:**

The manuscript delves into 3D medical image segmentation using a probabilistic approach. The approach comprises three integral components: generating probabilistic pseudo-labels, introducing a probabilistic multi-head self-attention mechanism, and formulating a probabilistic segmentation loss function. The proposed methodology has exhibited promising outcomes on the BTCV dataset.

**Strengths:**

1. The approach has demonstrated commendable results when evaluated on the BTCV dataset, showcasing its potential utility.
2. An advantage of the proposed method is its reliance on annotations that are more resource-efficient than exhaustive voxel-level annotations.
3. The paper stands out for its coherent organization and clear presentation of concepts.

**Weaknesses:**

1. The paper's technical contribution seems limited. Notably, the described Probabilistic Multi-head Self-Attention (PMSA) mirrors the design presented by Guo et al. (2022). Moreover, leveraging probability for modulating the Cross Entropy loss isn't a novel endeavor in the domain.
2. A glaring omission is the absence of comparative analyses with established weakly-supervised semantic segmentation techniques, especially those harnessing Class Activation Mapping (CAM).
3. Regarding the labeling technique, the paper utilizes points evenly spread over organ surfaces. This approach begs several questions: Does this method necessitate medical practitioners to outline organ boundaries precisely? Does this annotation technique offer tangible time-saving benefits over traditional voxel-level labeling? If the chosen points were scattered randomly within organ interiors, would the model's efficacy plummet?

**Questions:**

1. The introductory segment of the paper makes a contentious claim: "Typically, these methods directly employ sparse weak annotations, such as points or scribbles, during the training phase, potentially resulting in significant information loss." This stance contradicts prevalent practices. Rarely do researchers utilize weak labels for directly supervising the end segmentation model. A more widespread strategy involves harnessing deep learning tools to craft pseudo-labels, which subsequently guide a fully supervised semantic segmentation model's training.
2. In the experiment that examines the effect of varying annotation point numbers (n), a puzzling observation emerges. As 'n' escalates, the model's performance doesn't see a proportional upswing. Strikingly, the model's output seems subpar for n = 150. This finding contradicts intuitive expectations. Can the authors shed light on this perplexing behavior?

---

### Official Review · Reviewer_Red3 · 2023-11-01

**Soundness:** 2 fair
**Presentation:** 2 fair
**Contribution:** 2 fair
**Rating:** 3
**Confidence:** 3

**Summary:**

This work proposes a probability-based pseudo label generation scheme which synthesize dense 3D segmentation masks from sparsely annotated point annotations. Then, the proposed self-attention network extract the probability-driven features, and serve as the basis of the probability informed segmentation loss.

**Strengths:**

(1)This work is aimed to solve the interesting problem of weakly-supervised 3D segmentation.
(2)The manuscript is well wirtten and clearly organized.

**Weaknesses:**

(1) Only evenly distributed annotations are evaluated, which is unavailabel in practice. The proposed method tackles the sparsely annotated points. As described in Sec 3.2.1., the annotators are instructed to select 3D points **evenly distributed on the surface of the targe organ**. However, in practice, the weak annotations exist in hospital systems might be distributed irregularly. If the proposed method is only evaluated on these evenly distributed annotations, it might lost the generalization ability to the datasets in real applications.

(2) The proposed confidence score also relies on the even distribution of point annotations, as it is based on the distance between the target point and the annotated point. Then, its performance could decrease significantly when adapted to the irregularly annotations.

(3) The proposed segmentation loss is a simple combination of Dice and PCE, to take the pseudo labels as supervision. However, this mechanism highly relies on the threshold (hyper-parameter) to determine which pseudo label should be taken. When the annotation is not ideal, there could be a lot of noise in the generated pseudo labels, and this pipline might lost its advantage.

(4) It lacks the comparison with SOTA weakly supervised medical image segmentation methods, such as  point-supervised segmentation, scribble-supervised segmentation, et al.

**Questions:**

1. This method is proposed to tackle the 3D medical image segmentation. Can it be adapted to 2D segmentation tasks?

2. For Sec 3.4, the KL divergence loss is introduced to align the distribution of dependency score and the manually-set prior distribution. How to determine this prior distribution and what is the difference between different priors?

3. My main concern is the reliance of the proposed method on uniformly distributed annotations.How does performance change if annotations have a biased distribution?

4. What's the annotation cost of proposed evenly distributed points? What is the difference between this annotation pattern and scribbles/randomly distributed points ?

---

### Official Review · Reviewer_Nq7G · 2023-11-03

**Soundness:** 1 poor
**Presentation:** 3 good
**Contribution:** 2 fair
**Rating:** 3
**Confidence:** 4

**Summary:**

This paper proposes a weakly supervised 3D medical segmentation network from the probabilistic approach. The method incorporates annotation confidence and employs a probabilistic transformer network with a probability-informed segmentation loss function. Compared to the other weakly supervised methods presented in the paper, the proposed method provides better segmentation performance on the BTCV dataset.

**Strengths:**

- The proposed methodology looks reasonable. It makes sense to use probabilistic-aware learning in a weakly supervised settings for 3D medical image segmentation which could reduce the annotation cost.
- The paper leverages recent powerful techniques (e.g., UNETR, etc..) and aims to improve with a Gaussian-based multi-head self-attention mechanism.
- The paper provides ablation study and comparisons with the existing methods with a known public dataset.
- The paper is well-written and easy-to-read.

**Weaknesses:**

- The major weakness is the evaluation. I think that the experiments are simulated and may not be reflective of the real world. In the process of sparse label generation, the sampling algorithm produces evenly distributed points as annotations, but this may not be realistic and practical. It is possible that the annotation generation process have a major impact on the overall performance increase.
- Another problem with the evaluation is that the results in the paper come from the model's performance on the validation dataset, not a hidden testing dataset. The model and hyperparameters are optimized on the validation dataset so I am not sure if the results can be generalized, even on a dataset from the same distribution.
- The number of sample points is important as shown by the paper. But the variations in DICE and HD metrics raise questions about the stability of the proposed method. For example, why does the performance significantly drop when the number of points changes from 200 to 150?
- The paper defines the overall probability-aware segmentation loss function as L_total = L_Seg + wL_KL. The balancing term w is set to 0.3 but the author does not explain how this is determined in the paper. This may be highly critical and I am wondering how the stability and the performance is affected by the change of this parameter.
- I highly encourage the authors to provide some details about the overall complexity of their proposed method and its comparison with the existing weakly supervised methods in the literature.

**Questions:**

I look forward to seeing the authors responses for the concerns in the weakness section..